# Occurrence of *Cryptosporidium* spp. and *Giardia* spp. Infection in Humans in Latvia: Evidence of Underdiagnosed and Underreported Cases

**DOI:** 10.3390/medicina58040471

**Published:** 2022-03-24

**Authors:** Gunita Deksne, Agris Krūmiņš, Maira Mateusa, Vladimirs Morozovs, Dārta Paula Šveisberga, Rita Korotinska, Antra Bormane, Ludmila Vīksna, Angelika Krūmiņa

**Affiliations:** 1Institute of Food Safety, Animal Health and Environment BIOR, 1076 Riga, Latvia; maira.mateusa@bior.lv (M.M.); darta.sveisberga@bior.lv (D.P.Š.); angelika.krumina@rsu.lv (A.K.); 2Faculty of Biology, University of Latvia, 1004 Riga, Latvia; 3Post-Graduate Study Program in Family Medicine, University of Latvia, 1004 Riga, Latvia; agris.krumins@inbox.lv; 4Faculty of Veterinary Medicine, University of Life Sciences and Technology, 3001 Jelgava, Latvia; 5Faculty of Medicine, Riga Stradiņš University, 1007 Riga, Latvia; vmwork44@gmail.com (V.M.); ludmila.viksna@rsu.lv (L.V.); 6Centre for Disease Prevention and Control of Latvia, 1005 Riga, Latvia; rita.korotinska@spkc.gov.lv (R.K.); antra.bormane@spkc.gov.lv (A.B.)

**Keywords:** cryptosporidiosis, giardiasis, children, underdiagnosed, the Baltic states

## Abstract

*Background and Objectives*: Protozoan parasites—*Cryptosporidium* and *Giardia*—are important causes of diarrhea with an underestimated short-term burden on childhood growth and wellbeing in children under five years of age. The main transmission routes for both parasites are food and drinking water; transmission from person to person; and, due to their zoonotic nature, from domestic or wild animals to humans. The aims of the present study were to summarize the officially reported human cases of cryptosporidiosis and giardiasis in Latvia and to assess the occurrence of *Cryptosporidium* and *Giardia* in children within a prospective prevalence study. *Materials and Methods*: The number of officially reported cases of cryptosporidiosis and giardiasis in the time period of 2000–2020 was collected from the Centre for Disease Prevention and Control of Latvia. Data from a clinical diagnostic laboratory were included in the study in the period from 1 January 2008 to 31 December 2018. Additionally, a prospective study was performed, and fecal samples were collected from unique 0–17-year-old patients from January to February 2021 and tested using fluorescent microscopy. *Results*: Overall, during the 20-year period, 71 cases (mean per year = 9) of cryptosporidiosis and 1020 (mean per year = 34) cases of giardiasis were officially reported in Latvia. Meanwhile, within the prospective study, we found 35 (6.0%; 95%CI 4.3–8.1) *Cryptosporidium* and 42 (7.2%; 95%CI 5.3–9.6) *Giardia* cases. *Conclusions*: Here, we provide clear proof that both *Cryptosporidium* and *Giardia* are underdiagnosed in Latvia, which could also be true for neighboring Baltic and European countries, where a low number of cases are officially reported. Therefore, we highlight the hypothesis that the actual number of cryptosporidiosis and giardiasis human cases in the Baltic states is higher than that officially reported, including in Latvia.

## 1. Introduction

Protozoan parasites are important causes of diarrhea and other enteric diseases in humans [1]. *Cryptosporidium* spp. are among the leading causes of moderate-to-severe diarrhea, with an underestimated short-term burden on childhood growth and wellbeing in children under five years of age [2]. Likewise, *Giardia duodenalis* infects approximately 200 million individuals worldwide and causes acute diarrhea in children under 5 years of age [3]. The epidemiology of cryptosporidiosis and giardiasis is complex. The main transmission routes for *Cryptosporidium* and *Giardia* are food and drinking water, but transmission from person to person and, due to the zoonotic nature of these protozoans, from domestic or wild animals to humans may also occur [4]. Foodborne parasites are recognized as significant foodborne pathogens, but they still remain neglected compared with bacterial and viral pathogens. The World Health Organization has included both parasites in the “Neglected Disease Initiative” list since 2004 [5]. Meanwhile, *Cryptosporidium* spp. and *G. duodenalis* were ranked as the fifth and seventh most important foodborne parasite in Europe, respectively [6].

The genus *Cryptosporidium* consists of more than 40 species and an equal number of *Cryptosporidium* genotypes of unknown species status [7,8,9]. These species have several mammalian and non-mammalian hosts, and cross-infections may occur between various host species [10]. The genus *Giardia* comprises seven species, including *Giardia duodenalis* (also referred to as *G. intestinalis* or *G. lamblia*), which can infect several hosts, including mammals [11,12]. *G. duodenalis* is grouped into eight assemblages (from A to H), of which only A and B are infectious to humans, but it can also infect livestock and companion and wild animals, which, in turn, makes *Giardia* a zoonotic pathogen [13]. The main risk groups for cryptosporidiosis are children and immunocompromised people (e.g., patients with untreated Human immunodeficiency virus (HIV) infection, Acquired immune deficiency syndrome (AIDS), or cancer and transplant patients). Infection in malnourished children—especially children younger than two years of age—is strongly correlated with cognitive dysfunction, stunted physical growth, poor physical fitness, and high mortality [14,15]. Meanwhile, an increased risk for an infection with giardiasis has been associated with immunodeficiency, hypogammaglobulinemia, and isolated immunoglobulin A deficiency [16].

The prevalence of *Cryptosporidium* and *Giardia* infection in humans in Nordic countries (Finland, Norway, Sweden, and Denmark) varies between 0.0 and 13.6% for asymptomatic cases and 0.9–10.9% for symptomatic cases of *Cryptosporidium*, and 2.1–25.8% for asymptomatic cases and 12.6–47.4% for symptomatic cases of *Giardia* [17]. Both parasites are prevalent in Eastern Europe, but the number of reported cases varies greatly between the investigated countries; the causes of this variation include true differences in exposure and susceptibility; variable provision and access to healthcare systems; and differences in case definition, laboratory diagnosis, the recording of cases, and reporting. The national health systems of Eastern European countries operate differently [18].

Cryptosporidiosis in humans is a notifiable, but most likely underreported and underdiagnosed, disease in the official registers, and surveillance data provide a scarce overview of the epidemiology of cryptosporidiosis and giardiasis in Latvia [18,19]. The potential reasons for underestimation are that not all patients have severe symptoms and not all patients seek medical care. The symptoms are similar to those caused by bacterial or viral agents and can be misinterpreted; therefore, patients are less likely to be suspected of having *Cryptosporidium* and/or *Giardia* infection. Laboratory analysis might fail to detect these parasites because of the low number of excreted (oo)cysts in feces and the inappropriate methods used for routine diagnostics [20]. Thus, subclinical, asymptomatic, and undetected cases may play significant roles in infection transmission and epidemiology in the general population.

The aims of the present study were to summarize the officially reported human cases of cryptosporidiosis and giardiasis in Latvia and to assess the occurrence of *Cryptosporidium* and *Giardia* in children within a prospective prevalence study.

## 2. Materials and Methods

### 2.1. Study Design

The number of officially reported cases of cryptosporidiosis and giardiasis in the time period of 2000–2020 was collected from the Centre for Disease Prevention and Control of Latvia (CDPC) [21]. Additionally, data from a clinical diagnostic laboratory were included in the study, with only unique patients tested for the presence of *Cryptosporidium* and *Giardia* with the appropriate methods in the period from 1 January 2008 to 31 December 2018. The use of this data to characterize the epidemiology of an infectious disease presents a unique opportunity to assess the prevalence of infection with minimal cost while allowing for the characterization of occurrence across time. This approach also facilitates the identification of any patterns that this infection might have with respect to patient age, gender, and common comorbidities.

For the prospective study, fecal samples were collected from a clinical diagnostic laboratory in Latvia from unique 0–17-year-old patients. To find out how many children are infected with *Cryptosporidium* and *Giardia* in Latvia, we calculated the minimal sample size with OpenEpi v.3.01 [22], assuming that 50% of children are infected to obtain the maximum variability with a 95% confidence level. This established that a minimum of 385 random samples should be collected. Sampling was carried out from January to February 2021, and samples were stored at −20 °C before further analyses. Data about gender, age, and prior diagnostic purposes were collected. Overall, 641 fecal samples were collected.

This study was conducted with the approval of Riga Stradinš University Ethics committee (No. 6-1/13/2020/51).

### 2.2. Fluorescent Microscopy Analyses

Samples were thawed overnight at +4 °C before further analyses. For flotation with a saturated NaCl solution (relative density 1.18–1.2), 1 g of fecal sample was used, and after the flotation and centrifugation steps, 2 mL of purified material was available for subsequent analyses. For the fluorescent microscopy, 10 μL of the thoroughly suspended purified oocysts was added to the well of a Teflon slide (i.e., a Teflon slide with three 12 mm wells; Immuno-Cell, Mechelen, Germany), dried on the well, and fixed by submerging the slide in methanol for five minutes. The material was stained with FITC-labeled anti-*Cryptosporidium*/*Giardia* mAbs (AquaGlo, Waterborne, Inc., New Orleans, LA, USA), and further steps were carried out following the manufacturer’s instructions. For enumeration, brightly stained (oo)cysts with typical morphology were counted in all wells at 200× magnification. Each detected (oo)cyst represents 200 OPG.

### 2.3. Data Analyses

Patient age was divided into nine groups: <1 year; 1–6 years; 7–14 years; 15–17 years; 18–24 years; 30–39 years; 40–49 years; 50–59 years; and >60 years.

To calculate the confidence limits for point estimate proportions (e.g., proportion of *Cryptosporidium-* and *Giardia*-infected patients per age group and per different sex), both in retrospective and prospective studies, we assumed a binomial distribution, and 95% confidence intervals were calculated using the Mid-p Exact of the open-source software OpenEpi v.2.3.1 [22]. Two-tailed *p* < 0.05 was considered statistically significant.

## 3. Results

### 3.1. Long-Term Data on Officially Reported Cryptosporidiosis and Giardiasis Cases in Latvia

Overall, during the 20-year period, 71 cases of cryptosporidiosis and 1020 cases of giardiasis were reported to CDPC. A significantly higher (*p* < 0.05) proportion of cryptosporidiosis cases were reported in the age group of 30–39 years (33.8%; 95%CI 31.7–57.8) (Figure 1). However, in the case of giardiasis, a significantly higher (*p* < 0.05) proportion of cases were reported in the age groups of 1–6 years (9.0; 95%CI 7.4–11.0) and 7–14 years (14.1; 95%CI 12.1–16.4) from all reported cases (Figure 2).

In total, data from 18.367 unique patients, who were tested for *Giardia* antigen presence from 2008 to 2018 in a clinical diagnostic laboratory, were included in the present study. Patient age ranged from 0 to 96 years (mean = 30.7, median = 31.0), and 60.4% (n = 11.097) of them were females and 38.4% (n = 7056) were males. Some data were missing regarding the gender of the patients. The overall proportion of *Giardia*-infected patients was 2.2% (95%CI 2.0–2.4), and there were no significant differences between the age groups. A significantly higher (*p* = 0.02) proportion of females (1.9%; 95%CI 1.6–2.1) were infected with *Giardia* compared to males (2.4%; 95%CI 2.0–2.8). The number of diagnosed cases per year in this laboratory made up a range from 11.1 to 86.4% of officially reported cases per year to CDPC, with the exception of the year 2012, when more *Giardia* cases (n = 22) were diagnosed than officially reported to CDPC (n = 17). Meanwhile, during the time period 2008–2018, there were no samples tested for the presence of *Cryptosporidium* in this specific clinical diagnostic laboratory.

### 3.2. Cryptosporidium/Giardia Presence in Children within the Prospective Study

Overall, 584 samples were suitable for analyses. Patients ranged in age from 22 days to 17 years (mean = 4.6 and median = 3.0), and 45.9% (n = 268) were females and 54.1% (n = 316) were males. The fecal samples were submitted to the clinical laboratory for the following analyses: coprogram (79.1%), endoparasite detection with a concentration technique (42.6%), different bacteriological analyses (4.8%), calprotectin (1.5%), and occult blood (0.7%) for a single analysis or different analysis combinations.

There was no significant difference (*p* = 0.2) between the proportion of *Cryptosporidium* oocysts (6.0%; 95%CI 4.3–8.1) and *Giardia* cysts (7.2%; 95%CI 5.3–9.6) in the infected children. No significant (*p* > 0.05) differences were observed between the proportion of infected children with *Cryptosporidium* in different age groups (Table 1). Meanwhile, a significantly (*p* = 0.04) lower proportion of *Giardia*-infected children were observed in the age group of less than one year, and a significantly higher proportion (*p* = 0.04) were observed in the age group of 15–17 years (Table 1). There were no significant differences in the proportion of infected children between sexes for both parasites.

Infection with both parasites was observed in five children (1.3%; 95%CI 0.5–2.9): two were males and three were females; three were from the age group of 1–6 years, one was from the age group of 7–14 years, and one was from the age group of 15–17 years.

## 4. Discussion

Here, we provide clear proof that both *Cryptosporidium* and *Giardia* are underdiagnosed in Latvia. Within the present prospective study in children alone, we found 35 *Cryptosporidium* and 42 *Giardia* cases, while in the year 2021 (from January to November), there were 2 and 35 cases officially reported in Latvia, respectively [21] (www.spkc.gov.lv). Similar results could be found in neighboring Baltic and European countries, where a low number of cases are officially reported, despite *Cryptosporidium* and *Giardia* being a notifiable disease in humans [18,19]. In the year 2020, there were 5 officially reported cases of cryptosporidiosis and 86 cases of giardiasis in Estonia; there were no officially reported cases of cryptosporidiosis and 11 cases of giardiasis in Lithuania [23].

In most countries, surveillance in humans is passive, and countries only report diagnosed patients with clinical symptoms or hospitalized cases [19]. In Latvia, hospitalized patients, both adults and children, expressing gastrointestinal symptoms are primarily tested for viral (rotavirus, adenovirus, norovirus, etc.) and bacterial pathogens (*Salmonella*, *Shigella*, *Campylobacter*, *Yersinia*, etc.). Testing is initiated for parasites, including *Cryptosporidium* and *Giardia*, only when viral and bacterial pathogens are excluded. According to the handbook available for Latvian physicians, *Cryptosporidium* and *Giardia* should be considered in the following conditions: only in cases of chronic diarrhea (longer than 14 days); in cases of immunosuppressed patients with HIV/AIDS; and in cases where the patient has recently travelled to a country with a warmer climate [24]. Unlike *Cryptosporidium*, testing is state funded for the previously mentioned viruses, bacteria, and *Giardia* (National Health Service, www.vmnvd.gov.lv). Similar conditions for the primary testing of bacteria and viruses are mentioned in Finland [25].

One of the predisposing factors for the spread of *Cryptosporidium* and *Giardia* is the climate in the Baltic region, with its frequent rain and floods. Observations have provided an estimate (for the past 30 years) of mean annual precipitation of 750 mm/year for the entire Baltic Sea basin, including both land and sea [26]. A humid environment could lead to the massive spread of *Cryptosporidium* and *Giardia* [27].

*Cryptosporidium* and *Giardia* are highly successful parasites due to their large host range, high (oo)cyst output from infected individuals, waterborne and foodborne transmission routes, and low infectious dose. The latest modeling studies have shown a significant risk of infection with *Cryptosporidium* from as low as 1 oocyst, while a *Giardia* infectious dose can be as low as 10 cysts [28,29]. Due to their capacity for survival in the environment and zoonotic dispersal, these parasites can pose a significant risk to public health, especially through contaminated drinking water and public swimming pools.

Data on the epidemiology and the zoonotic potential of *Cryptosporidium* and *Giardia* shed by animals and circulating in the Baltic states, including Latvia, are scarce. Surveillance data in livestock show that *Cryptosporidium* spp. herd prevalence ranges from 66% in Estonia to 72.4% in Latvia and 100% in Lithuania [30,31,32]. There are limited studies of the presence of *Cryptosporidium* and *Giardia* in other animal species [18]. The results in the present study provide evidence that *Cryptosporidium* and *Giardia* are widely present in the environment of the Baltic states. *Cryptosporidium* oocysts are robust and extremely resistant to chemical disinfection (chlorine), and oocysts can also survive for several months in water and soil [33,34], while *Giardia* is more susceptible to environmental conditions [35]. Based on this, we highlight the hypothesis that the actual number of cryptosporidiosis and giardiasis human cases in the Baltic states is higher than that officially reported, including in Latvia.

We provide data that estimate the importance and the risk to public health caused by *Cryptosporidium* and *Giardia* in Latvia. Both parasites can cause significant morbidity with low infective doses, even for fully immunocompetent persons, but especially for young, old, pregnant, and immunocompromised persons [10]. The spread of the pathogens in children is due to poor hygiene and a lack of understanding of hygiene, as children tend to rarely wash their hands appropriately after restroom use and before meals [36].

## 5. Conclusions

Cryptosporidiosis and giardiasis are considered as possible causal agents in the case of long-lasting watery diarrhea combined with abdominal cramps, and fecal samples should be tested for *Cryptosporidium* and *Giardia* at the same time as when testing is performed for bacteria and viruses. We still lack information regarding infection routes and animal hosts; therefore, further studies on these pathogens will be conducted to better understand the epidemiology and parasite diversity in Latvia. Further parasite prevalence studies in animal populations and molecular epidemiology studies will improve the knowledge of the main *Cryptosporidium* and *Giardia* transmission routes from animals (such as pets, wild animals, and livestock) through the environment (water, sewage, and soil) to humans.

## Figures and Tables

**Figure 1 medicina-58-00471-f001:**
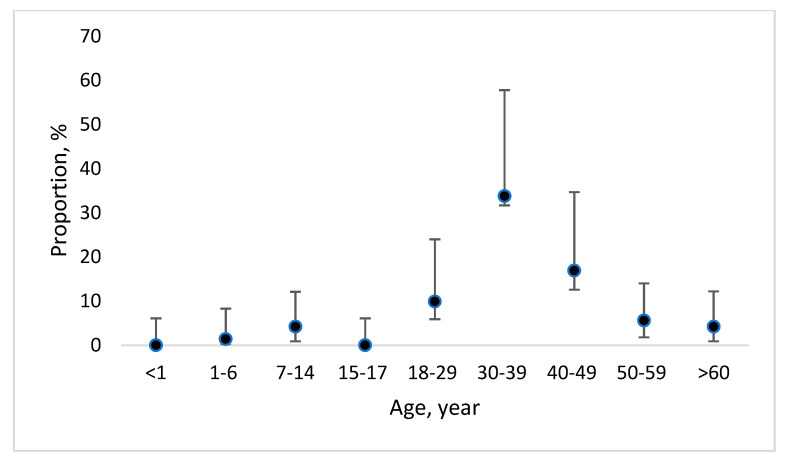
Proportion with 95% confidence interval of officially reported cryptosporidiosis cases per age group over the time period from 2000 to 2020 in Latvia.

**Figure 2 medicina-58-00471-f002:**
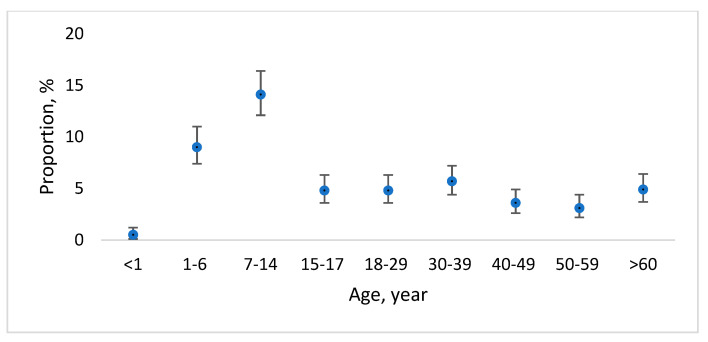
Proportion with 95% confidence interval of officially reported giardiasis cases per age group over the time period from 2000 to 2020 in Latvia.

**Table 1 medicina-58-00471-t001:** Number of analyzed, infected, and proportion with 95% confidence interval (95%CI) of children with *Cryptosporidium* spp. and *Giardia* spp. in different age and sex groups.

Factor		Not Analyzed	*Cryptosporidium* spp.		*Giardia* spp.	
			No. of Infected	Proportion, %	No. of Infected	Proportion, %
				(95% CI)		(95% CI)
Age group	<1	123	4	3.3 (1.0–8.3)	3	2.4 (0.5–7.2)
	1–6	313	22	7.0 (4.6–10.5)	23	7.3 (4.9–10.8)
	6–14	108	6	5.6 (2.3–11.8)	8	7.4 (3.6–14.1)
	15–17	40	3	7.5 (1.9–20.6)	8	20 (10.2–35.0)
Gender	Female	268	11	4.1 (2.2–7.3)	23	8.6 (5.7–12.6)
	Male	316	24	7.6 (5.1–11.1)	19	6.0 (3.8–9.3)
Total		584	35	6.0 (4.3–8.1)	42	7.2 (5.3–9.6)

## Data Availability

The analyzed data sets supporting the results are partly available on www.spkc.gov.lv (accessed on 20 January 2022).

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
