# Peer review of "Occurrence of Cryptosporidium spp. and Giardia spp. Infection in Humans in Latvia: Evidence of Underdiagnosed and Underreported Cases"

_medicina, 2022, doi:10.3390/medicina58040471_

Round 1

Reviewer 1 Report

General Comments: 

The objectives of this research was to study the prevalence of Giardia and Cryptosporidium in Latvia. Generally the manuscript is well written with a few spelling errors. The aims and objectives are covered in the manuscript and this information is essential especially when it comes to neglected diseases. However, there are major additions to be made to the methodology in order for the paper to meet publication standards

Specific comments: 

Abstract: 

The prevalence of Cryptosporidium and Giardia could have been mentioned with faecal samples taken from fluorescent microscopy with patients aged 0-17 years. How it is presented may cause confusion.

Introduction: 

Line 78: "Symptoms are similar to bacterial or viral agents caused symptoms" Please reword.

Line 79: "fecal samples less likely would be analyzed for the presence" Please reword.

Methodology: 

Sample size calculations can be included in this section. What was the calculated sample size in comparison to the number of samples obtained in the study. This should be referenced. (Sample size for detection of disease; approx. prevalence; total population; confidence interval). This information is required if work has to be published.

More information is needed for data analysis. What tests was performed with various data (prospective and retrospective data). eg Chi Square test for the population data

Results: 

The data is well presented as is.

Discussion:

Line 209: Italicize Giardia, please check the entire manuscript to ensure this is done.

Conclusion:

This is well written and agrees with what was presented in the manuscript.

In closing: Adding a section on recommendation/ further work will add tremendously to the manuscript. It can state research in wild animal species to identify these parasites as well as testing of water sources (pools, tanks etc) or food. 

Author Response

Responses to Rewiever 1 points:

Authors response: We are grateful for the suggestions and pertinent comments. The manuscript has gone through revision considering reviewers recommendations. During the revision, we improved the necessary explanations. 

Specific comments: 

Abstract: 

R1: The prevalence of Cryptosporidium and Giardia could have been mentioned with fecal samples taken from fluorescent microscopy with patients aged 0-17 years. How it is presented may cause confusion.

Author response: Thank you for your comment. We have corrected the sentence to make the results more clear.

Introduction: 

R1: Line 78: "Symptoms are similar to bacterial or viral agents caused symptoms" Please reword.

AR: Rewritten

R1: Line 79: "fecal samples less likely would be analyzed for the presence" Please reword.

AR: Rewritten

Methodology: 

R1: Sample size calculations can be included in this section. What was the calculated sample size in comparison to the number of samples obtained in the study. This should be referenced. (Sample size for detection of disease; approx. prevalence; total population; confidence interval). This information is required if work has to be published.

AR: Thank you for your remark! We have improved the Material section by adding the description of sample size calculation as well as the number of collected samples.

R1:More information is needed for data analysis. What tests were performed with various data (prospective and retrospective data). eg Chi Square test for the population data

AR: During the revision, we have added the clarification that for both, retrospective and prospective) studies, that for calculating confidence limits for point estimated proportions (e.g. proportion of Cryptosporidium and Giardia infected patients per age groups, per different sex), both in retrospective and prospective study, we assumed a binomial distribution and 95% confidence intervals were calculated using the Mid-p Exact of the open source software OpenEpi v.2.3.1. Two-tailed p < 0.05 was considered statistically significant.

Results: 

R1: The data is well presented as-is.

AR: Thank you for your opinion.

Discussion:

Line 209: Italicize Giardia, please check the entire manuscript to ensure this is done.

AR: Thank you for this remark. We have gone through the whole text carefully and checked both Cryptosporidium and Giardia and italicized when needed.

Conclusion:

R1: This is well written and agrees with what was presented in the manuscript. In closing: Adding a section on recommendation/ further work will add tremendously to the manuscript. It can state research in wild animal species to identify these parasites as well as testing of water sources (pools, tanks etc) or food. 

AR: Thank you for this suggestion. During the revision, we have improved the Conclusions by adding the explanations of present study limitations and adding the recommended further studies to improve knowledge of parasite transmission routes.

Reviewer 2 Report

This study was aimed to summarize the official reported human cases of cryptosporidiosis and giardiasis in Latvia and to assess the occurrence of Cryptosporidium and Giardia in children within a prospective prevalence study. In the paper, the materials and methods are efficiently described, and the conclusions are compatible with the results. This manuscript is found suitable for publishing in the Medicina.

Author Response

R2: This study was aimed to summarize the official reported human cases of cryptosporidiosis and giardiasis in Latvia and to assess the occurrence of Cryptosporidium and Giardia in children within a prospective prevalence study. In the paper, the materials and methods are efficiently described, and the conclusions are compatible with the results. This manuscript is found suitable for publishing in the Medicina.

Response: Thank you for your valuable opinion. During the revision we have improved the Method description and Conclusions.

Reviewer 3 Report

The Manuscript is interesting for the reader especially in the Baltic region. However, there were several basic defects in the methods such as

  1. It is not clear about the sample size in each catagories since you not mention the number of sample in the method section but then you mentioned it in the result section so maybe is will be more clear to mention that on methods section as well.
  2. It might be better to compare the results from each section to show the evidence of underestimate diagnosis and it will be more easy to understand.
  3. The conclusion should be more explanation including limitation of this study, specific further study that you recommended to do to better understand the real situation in Latvia or Blatic region for example.

Author Response

1. It is not clear about the sample size in each catagories since you not mention the number of sample in the method section but then you mentioned it in the result section so maybe is will be more clear to mention that on methods section as well.

Respons: Thank you for your remark! We have improved the Material section and included the description of sample size calculation as well as the number of collected samples.

2. It might be better to compare the results from each section to show the evidence of underestimate diagnosis and it will be more easy to understand.

Response: Thank you for your comment.

3. The conclusion should be more explanation including limitation of this study, specific further study that you recommended to do to better understand the real situation in Latvia or Blatic region for example.

Response: Thank you for this suggestion. During the revision, we have improved the Conclusions by adding the explanations of present study limitations and adding the recommended further studies to improve knowledge of parasite transmission routes.

Round 2

Reviewer 1 Report

The revisions that were made are appropriate.